# Study on Fabrication and Properties of Graphite/Al Composites by Hot Isostatic Pressing-Rolling Process

**DOI:** 10.3390/ma14102522

**Published:** 2021-05-12

**Authors:** Hao Jia, Jianzhong Fan, Yanqiang Liu, Yuehong Zhao, Junhui Nie, Shaohua Wei

**Affiliations:** 1GRINM Metal Composites Technology Co. Ltd., Beijing 101407, China; chinsland@sina.com (H.J.); 18612186108@163.com (Y.Z.); niejunkey@163.com (J.N.); weishaohua666@163.com (S.W.); 2General Research Institute for Non-Ferrous Metals, Beijing 100088, China; 3GRIMAT Engineering Institute Co., Ltd., Beijing 101407, China

**Keywords:** graphite film, metallic composites, thermal properties, rolling

## Abstract

Graphite/Al composites have attracted much attrition due to their excellent thermal properties. However, the improvement of thermal conductivity (TC) is limited by the difficulty in controlling the orientation of graphite and the poor wettability between graphite and aluminum. In this study, a novel process for fabricating the Graphite/Al composites is proposed, which involves fabricating graphite film and aluminum foil into laminate material. Then, taking a rolling method, the fractured and well oriented graphite film can help the composite achieve high TC while maintaining a certain strength. The result reveals that both single and total reduction have a significant influence on the diameter and orientation of the graphite, and by adjusting the process parameters, composites with high TC can be acquired at a relatively low reinforcement volume. This near-net-forming process can directly meet the thickness requirements for electronic packaging and avoids the exposure of graphite to the surface during secondary processing, which is promising to promote the application for high TC Graphite/Al composites in thermal management.

## 1. Introduction

With the rapid development of electronic devices towards miniaturization, lightweight, and high performance, the heat produced by the ever-increasing power density has become a key problem in restricting the reliability and efficiency of electronic equipment [1,2]. In order to effectively dissipate heat, the thermal management materials with high thermal conductivity (TC) and low coefficient of thermal expansion (CTE) which matches to the semiconductors are urgently needed. To date, thermal management materials have been developed for several periods, but some traditional thermal management materials, such as Invar, Kovar, SiC/Al, and Si/Al, can no longer fulfill the requirements for high power electronic devices due to insufficient TC [3,4]. In recent years, metal matrix composites (MMCs) reinforced by carbon materials, including carbon fibers [5,6], graphite, diamond [7,8], carbon nanotubes (CNTs) [9,10], and graphene [11], have become promising materials for thermal management applications due to their high TC and low density. 

In the previous study, some works have been done as a result to combine graphite with metal composite through different processes. Take graphite flake (Gf) as an example, Ouyang et al. [12] obtained the 50 vol% Gf reinforced composite by powder metallurgy, which had TC of 555 W/m·K. Li et al. [13] manufactured 70 vol% Gf/Al composites with preferred orientation and the TC reached 714 W/m·K. However, the flexural strength is less than 20 MPa, and due to the brittleness of graphite, increasing the volume fraction will inevitably lead to a decline in mechanical properties, so it is essential to achieve high TC with a relative low reinforcement volume so that the composite can maintain good mechanical properties.

In order to maximize the high TC of graphite materials and promote the application of Graphite/Al composite, there are still several problems that need to be solved. It is well known that poor wettability and interfacial reaction which produces Al_4_C_3_ between aluminum and graphite were harmful for the thermal stability, mechanical properties, and stacking of graphite leaves almost no space between the flakes, which make liquid metal difficult to infiltration in [14]. Besides, to take advantage of the anisotropic thermal properties of graphite, it is very important to control the orientation of graphite [15].

Recently, artificial graphite films fabricated from polyimide films have attracted much attention in thermal management applications due to their high graphitization degree and high TC [2,9]. The in-plane TC of graphite film can reach 1100–1600 W/m·k, which is much higher than that of graphite flake. Based on this, Huang et al. [16,17] used graphite films and aluminum foils as raw materials and fabricated laminate composites by vacuum hot pressing. Having successfully solved the problem on orientation, the TC of obtained composites can reach as high as 902 W/m·K. Nevertheless, its application was still restricted by low mechanical properties, and research shows that mechanical properties are closely related to the diameter of graphite [18]. When the load was applied, crack propagation more easily occurred on graphite to a great extent. In order to make up for the deficiency of laminated materials, the rolling process, which has been widely used in configuration controlling and interlayer bonding [19,20], was proposed in this study. And the SiC coating was grown on the surface of graphite film by molten salt method to improve its interfacial bonding with Al. The preparation sketch of the composites is shown in Figure 1. The fractured and well-oriented graphite film is helpful to maintain the high TC of the composites, meanwhile, the discontinuous structure can make full use of the strengthening effect of the metal matrix on the mechanical properties. The effects of process parameters on microstructure and properties of the composites were mainly investigated.

## 2. Materials and Methods

### 2.1. Raw Materials

Graphite films and pure Al 1060 foils (99.6% in purity, with TC of 234 W/m·K) were used as raw materials. Al foils with thickness of 50 μm ware purchased from Shanghai Yuanyi metals Co.Ltd (Shanghai, China). The graphite films with average thickness of 25 μm were acquired from Tanyuantech Co., Ltd. (Changzhou, China). The in-plane and out-of-plane TC of the graphite films were measured to be 1258 W/m·K and 18 W/m·K, respectively.

### 2.2. Preparation of the Composites

NaCl, KCl and Si powder (with the mass ratio of 4: 4: 1) was put into a mixing bowl and stirred in a two-dimensions mixer for 8 h at 200 rpm. Then the graphite films with mixed powder covered on the surface were set into a corundum crucible, and placed in the vacuum induction furnace, heated from 25 °C to 1150 °C in a vacuum induction furnace, and kept for 120 min in an argon atmosphere. After treatment, the coated graphite was cleaned in deionized water and dried at 100 °C continually. 

After coating, the graphite films and Al foils were cut into designed size, subsequently layered into laminates, then heated to 585 °C in a HIP furnace and kept for 120 min in 50 MPa. Afterward, the composites were processed by hot rolling. The roller diameter was 250 mm and the rolling speed was 40 rpm. Tempering occurred at 450 °C for 20 min per pass.

### 2.3. Characterization

The microstructure of the composites was observed by an optical microscope (OM, Axiovert A1, Zeiss, Oberkochen, Germany) and a field emission scanning electron microscope (FE-SEM, JSM-7600F, Jeol, Tokyo, Japan). Auger electron spectrometer (AES, PHI 710, Ulvac-Phi, Chigasaki, Japan) and energy dispersion spectra (EDS) attached to the FE-SEM were applied to determine and analyze the elemental composition of the coated graphite. The TC of composite λ was calculated by the following formula:(1)$$\lambda =\alpha \cdot C_{p}\cdot \rho$$
where α is the thermal diffusion coefficient of the composite measured at room temperature by a laser flash thermal analyzer (LFA-447, Netzsch, Selb, Germany). To meet the measurement requirements of thickness, the sample in size of 4 × 10 mm was stacked to about 10 mm. $C_{p}$ is specific heat of the composite, which was calculated according to rule of mixture:(2)$$C_{p}=C_{Al}\cdot V_{Al}+C_{Gr}V_{Gr}$$
where $C_{Al}$, $C_{Gr}$ is specific heat and $V_{Al}$, $V_{Gr}$ is the volume fraction of aluminum and graphite, respectively. Composite density was measured using the Archimedes principle.

## 3. Results

### 3.1. Microstructure of Graphite and Composites

The surface morphology and energy dispersion spectra (EDS) mapping result of Si-coated graphite are illustrated in Figure 2. The surface of coating was integrated and homogenous, and it can be seen from the EDS mapping results that the Si elements were evenly dispersed on the graphite. The EDS spectrum shows the O, Si, and C signals. The AES was applied to investigate the phase of the coating. As shown in Figure 2d, the valence state can be judged by the chemical shift of pattern. The Auger energy of silicon was measured to be 1608.3 eV, compared with Si (1618 eV), SiO_2_ (1605 eV) and Si_3_N_4_ (1612 eV), which indicates the surface silicon has charge of plus four. Moreover, the Auger energy of oxygen was measured to be 511.4 eV, compared with O_2_ (510 eV) and SiO_2_ (502 eV). The oxygen element on the surface probably exists as an impurity, and the composition of the coating can be determined to be SiC.

The coated graphite films and Al foils were layered into laminate, the volume of graphite was controlled at 11.1% and the initial thickness was 18 mm. Figure 3 shows the morphology of the composites after the HIP process, where we can see the graphite film was well aligned in the composites, as expected, and no voids appeared on the interface. Afterward, the composite was machined by hot rolling, and the effects of single and total reduction were mainly compared.

As is well known, Al has good ductility while graphite is a brittle material, the rolling process was aiming to let graphite fracture and keep parallel orientation when the composites extending. As shown in Figure 4, comparing the influence of different total reduction (a, b, c), it can be seen at the beginning of rolling, large initial thickness leads to large absolute reduction and bite angle, the deformation rate varies greatly in different regions along the height direction [21]. Thus, the main deformation modes of the composites are interlaminar extrusion and slipping. Some shear bands developed in the composite, and the fracture of graphite mainly occurs at the junction of shear bands. With the rolling processing, the reduction in thickness makes it easier for deformation to penetrate into the interior, and the deformation rate along the height direction tends to be uniform. Graphite begins to fracture along the rolling direction. Shear bands were flattened and new shear band no longer appears. Then, comparing the influence of different single reductions (b, d, e), with the single reduction increases, passes to reach the same total reduction were also reduced, and the difference of deformation rate will result in a heterogeneous structure, and some regions still maintain a layered structure.

Figure 5 shows the typical deformation modes of graphite during rolling. As a multi-layer material, the graphite film will also be affected and delaminated when the shear bands appear in the composites. From Figure 5a, we can see the graphite after delamination is distributed along the shear direction. When the heat flow in parallel direction passes through the composites, due to the low axial TC of graphite, this kind of structure will inevitably have adverse effects on the TC. Besides, when the graphite fractures along the parallel direction, the internal cracks will spread out unevenly, so the fracture edge is a spindle. The size of graphite varies with the fracture degree, so it is necessary to study the TC of the composites under different structures.

### 3.2. Modeling and Thermal Properties

To figure out the effect of graphite size and orientation on TC, three 12-mm samples for each process were selected and use Image-Pro Plus (IPP) to count the diameter of graphite. Besides, recent studies show that the inclination angle of the graphite will strongly weaken the in-plane TC, when the inclination angle of the reinforcement in the composite is greater than 15°, its in-plane TC will be lower than 90% of the theoretical value [15,18]. Therefore, the proportion of graphite with an inclination angle less than 15° was counted to evaluate the degree of alignment (grey area in Figure 6).

As shown in the figure, the *x*-axis and *y*-axis represent the inclination angle and diameter of graphite, respectively. With increasing in total reduction from 50% to 87.5%, the average diameter of graphite decreases from 791 μm to 228 μm accordingly. And with the graphite gradually tends to be parallel oriented, the proportion of the graphite with inclination angle within 15° increased from 67% to 88%. Furthermore, the influence of different single reductions was compared (b, d, e), and with single reduction increased from 15% to 30%, rolling passes to reach the same total reduction were reduced. Fewer rolling passes make the size of graphite vary greatly in different areas, and the average size is also larger than the composites prepared in lower single reduction. Simultaneously, a large single reduction will cause an uneven interlayer structure. The proportion of graphite with an inclination angle less than 15° decreased from 84% to 70%. 

To better understand the influence of graphite size on TC, a formula for calculating the effective TC was given by [22,23]: (3)$$K_{c}^{ii}=K_{s}\frac{K_{s}+L_{ii}\left(K_{c}-K_{s}\right)\left(1-\upsilon \right)+\upsilon \left(K_{c}-K_{s}\right)}{K_{s}+L_{ii}\left(K_{c}-K_{s}\right)\left(1-\upsilon \right)}$$
(4)$$\upsilon =\frac{a^{2}c}{{\left(a+\delta \right)}^{2}{\left(c+\delta \right)}^{2}}$$
where $K_{c}$ is the intrinsic TC of the graphite film, $K_{s}$ is the interfacial thermal conductance. δ is the thickness of the interface layer, while *a* and *c* are the short and long axes of reinforcement. In order to simply the calculation, the graphite was regarded as a disklike particle, $L_{ii}$ is the shape factor in different directions (for disklike particle, *L_x_* = *L_y_* = π*p*/4), considering the thin interface layer ($K_{s}$→0), and the in-plane effective TC of reinforcement was calculated in Figure 7a. It is obvious that the effective TC decreases significantly when the diameter of graphite is less than 250 μm, the small sized graphite is more likely to negatively affect the effective TC. Then, bringing the formula of effective TC into effective medium approach (EMA) model, the in-plane TC of composite can be expressed as:(5)$$K_{}^{*}=K_{m}(1+\frac{f}{\frac{\pi p}{4}\left(1-f\right)+\frac{K_{m}}{K_{c}^{ii}-K_{m}}})$$
where *f* is the volume fraction of graphite. *K_m_* is the TC of aluminum. Bring the average diameter and actual diameter which counted by software into the formula, the theoretical TC and the modified TC including size effects can be calculated respectively. As shown in Figure 7b, comparing the influence of different total reduction (a, b, c), with the rolling processing, the proportion of small-sized graphite with effective TC less than 80% (yellow area in Figure 4) was increased to 44%, resulting in the modified value to fall faster than the theoretical value. After excluding the influence of size, another main factor affecting the TC was the orientation of graphite. As the graphite tends to be parallel oriented, the gap between the experimental value and the modified value is also decreased. This phenomenon is especially obvious when comparing different single reductions (b, d, e). The average size of the graphite fabricated by 30% single reduction was larger than that of 15% single reduction, but the TC of the reinforcement has not been effectively played. The graphite with non-parallel orientation and the irregular structure have restricted the TC of composites, which increase the gap between the experimental value and the modified value.

### 3.3. Optimization of the Process

The analysis above shows the main factors influencing the TC are the size and orientation of graphite, large absolute reduction is likely to cause the uneven structure, while too many rolling passes will increase the proportion of small-sized graphite. In order to further study the influence of absolute reduction and thickness of rolled piece on metal flow during the rolling process, ignore the influence of the graphite, the internal metal flow of 40 mm and 20 mm thick aluminum plates under 30%/15% single reduction was simulated by Abaqus, the roller size and rolling speed are the same as the experiment, and the friction coefficient between roller and aluminum plates is set to 0.3.

As shown in Figure 8, when the rolled piece is thick, the large absolute reduction leads to large stress angle and long contact arc, which results in a large velocity gradient in the deformation zone, this may be the main reason for interlaminar shear deformation during rolling. Besides, the metal flow rate along the height direction is very uneven in the deformation area, but tends to be consistent in the exit area, which will also lead to excessive differences in the degree of deformation of the rolled piece. By reducing the single reduction to 15%, the rolled piece is mainly affected by the friction of the roller, the deformation cannot penetrate into the interior, and the metal flow rate between the surface layer and the inner layer is quite different. As the thickness of rolled piece decreases to 20 mm, with the decrease in absolute reduction and stress angle, the area of velocity gradient decreases correspondingly, further reducing the single reduction. Thus, the metal flow rate in different areas of the rolled piece tends to be uniform.

Therefore, an optimal scheme is proposed as follows: (1) By adding the extra layer on both sides of the rolled piece, so that the adverse effect caused by the difference of metal flow rate between the surface layer and the inner layer can be reduced; (2) Reduce the thickness of rolled piece.

The composite with initial thickness of 2 mm and 20% volume of graphite was prepared, and the total thickness was increased up to 8 mm, 12 mm, and 16 mm by adding the aluminum layer. The total reduction is 75% and the single reduction is 15%. After rolling, the surplus aluminum on the material surface was milled off. As shown in Figure 7. Obviously, the orientation of graphite was improved by reducing the thickness of rolled piece, however, the interlayer structure was still slightly affected by the velocity difference between surface layer and the inner layer, by further increasing the thickness of the additional layer to 16 mm, the proportion of graphite with inclination angle less than 15° can reach up to 95%. Furthermore, when the thickness of a rolled piece is 8 mm, the elongation is also very low under the same total reduction, which limits the fracture of graphite. The average size of graphite counted by IPP is 612 μm. With the increase in thickness, the average size of graphite was reduced to 557 μm and 439 μm, respectively. As shown in Figure 9d, after process optimization, the in-plane TC all approach or exceed 90% of the theoretical value, and mainly being affected by the orientation of graphite, when the whole graphite tends to be parallel oriented, the experimental value is 354 W/m·K, which can reach 97.8% of the modified value.

Figure 10 shows the tensile fracture morphology of composites with different total thickness. Normally, the strength of graphite is much lower than that of Al matrix, which leads to cracks preferentially generated on the graphite. In addition, affected by the different microstructure of the composites, the fracture also presents different characteristics. As shown in Figure 10a, the fracture surface is an inclined plane, and there is peeling between graphite and matrix in some areas. This is due to the shear band in the composite, when the composite was subjected to a horizontal load, the force will directly act on the interface between graphite and Al. Besides, the graphite film is composed of many graphite layers, the bonding between the layers is very weak, so the main failure mode of graphite is delamination, and the cracks will rapidly expend into the matrix, jagged tear can be observed in partial fracture region, the fracture surface is very uneven. Therefore, the tensile strength of the composites is also relatively lower.

When the graphite tends to be parallel orientated, the main failure mode of graphite is transformed into horizontal fracture. However, due to the large size of graphite, the effect of the Al matrix constraint is limited. As the average size of graphite further decreases, the crack propagation distance in the graphite can be more effectively constrained by the Al matrix. As shown in Figure 10c, the fracture surface is characterized as a layered distribution. The fracture of the graphite and the necking of the surrounding Al matrix can be observed, which indicates that the reinforcement effect of the matrix is effectively exerted, and the tensile strength of the composites was improved correspondingly. The properties of the composites prepared with different initial thickness are listed in Table 1.

Figure 11 shows the comparison of the in-plane TC of the composites fabricated in this study with that of other studies, in which we can see that the composites prepared in this study can achieve higher TC under the same graphite content. This is probably due to the higher intrinsic TC of graphite film compared with other carbon materials. Moreover, through optimizing the rolling process, the graphite can fracture and maintain a parallel orientation in the composite. This discontinuous configuration can make the composite keep a certain mechanical property. Moreover, the composite fabricated in this study can be as thin as 0.6 mm. This near-net-forming process can directly meet the thickness requirements for electronic packaging and avoids the exposure of graphite to the surface due to secondary processing, which is promising to provide a novel approach to promote the application for high TC C/Al composites.

## 4. Conclusions

In this study, a novel preparation process for Graphite/Al composites with high TC was proposed. This was achieved by fabricating the aluminum foil and graphite film with ultra-high TC into laminate material, then changing the composite into a discontinuous configuration via a rolling process. The result reveals that both single reduction and total reduction have a significant influence on the size and orientation of the graphite. Through process optimization, the fractured and well oriented graphite film can make the composite have high TC while maintain certain mechanical properties. When the volume fraction of graphite is 20%, the in-plane TC of the composite can reach as high as 354 W/m·K, while the tensile strength is 77 MPa. Moreover, it is expected that the properties of the composites can be further improved by adjusting the graphite volume and adopting an aluminum alloy matrix with higher strength.

## Figures and Tables

**Figure 1 materials-14-02522-f001:**
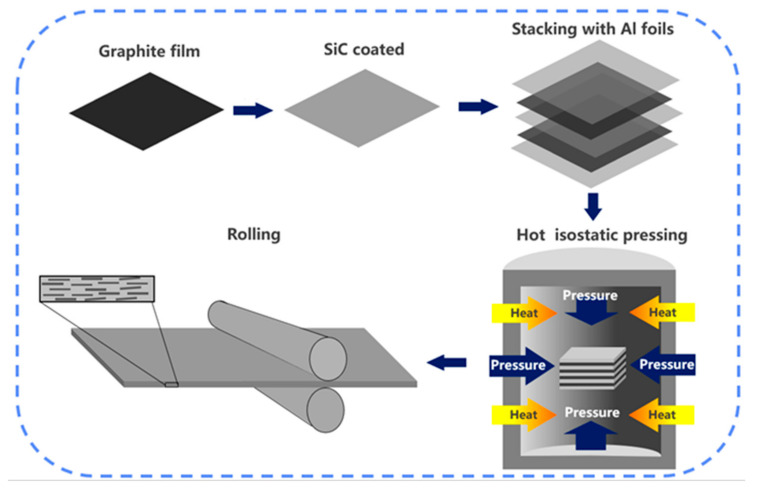
Fabrication process for graphite film/aluminum composites.

**Figure 2 materials-14-02522-f002:**
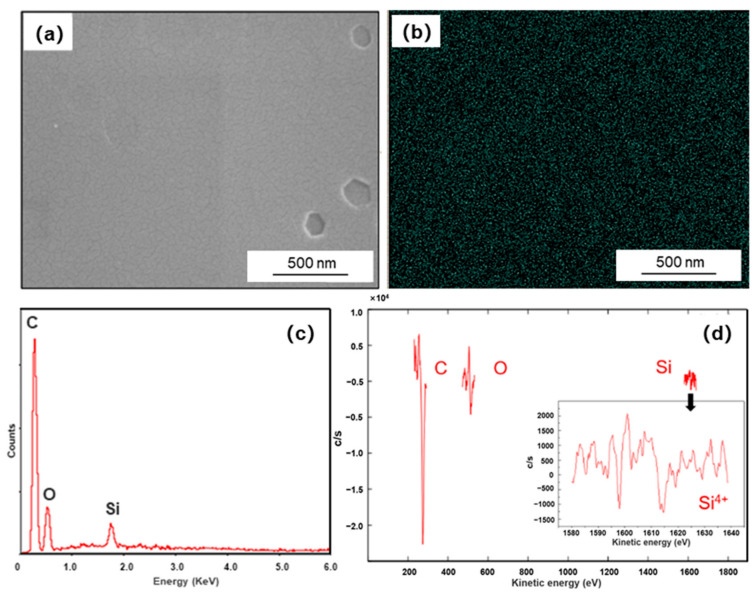
(**a**) SEM morphology of Si-coated graphite; (**b**) EDS mapping results of Si; (**c**,**d**) EDS spectrum and AES pattern of the Si-coated graphite.

**Figure 3 materials-14-02522-f003:**
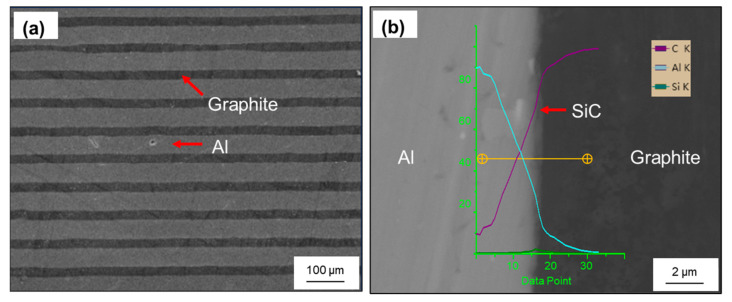
(**a**) SEM morphology of the composite after HIP process; (**b**) interfacial microstructure and EDS line-scan analysis.

**Figure 4 materials-14-02522-f004:**
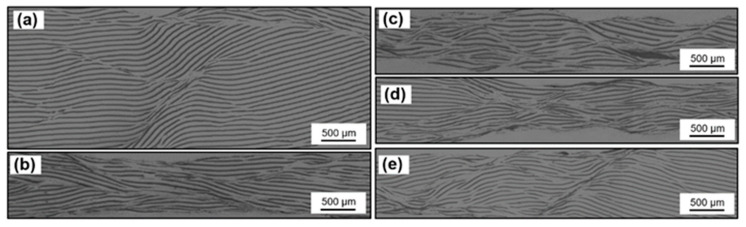
Morphology of composites with different single/total reduction (**a**) 15%/50%; (**b**) 15%/75%; (**c**) 15%/87.5%; (**d**) 20%/75%; (**e**) 30%/75%.

**Figure 5 materials-14-02522-f005:**
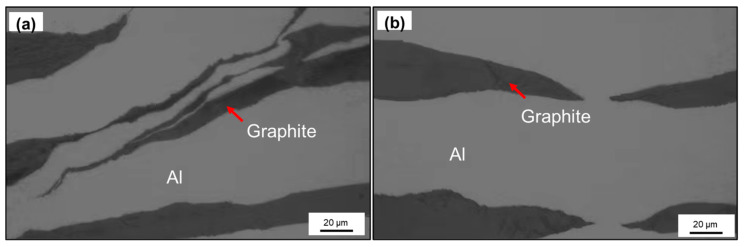
Typical deformation modes of graphite in rolling process: (**a**) slip, (**b**) fracture.

**Figure 6 materials-14-02522-f006:**
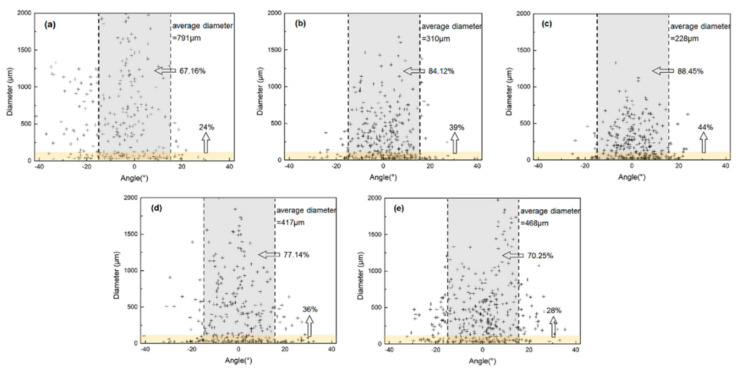
Inclination angle and size distribution of graphite in composite with different single/total reduction. (**a**) 15%/50%; (**b**) 15%/75%; (**c**) 15%/87.5%; (**d**) 20%/75%; (**e**) 30%/75%.

**Figure 7 materials-14-02522-f007:**
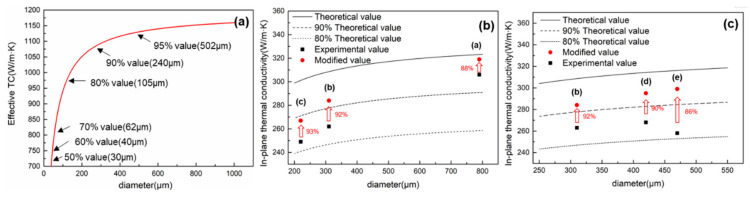
(**a**) Variety rule of the theoretical effective TC of graphite with diameter; (**b**,**c**) In-plane TC of composite with different single/total reduction.

**Figure 8 materials-14-02522-f008:**
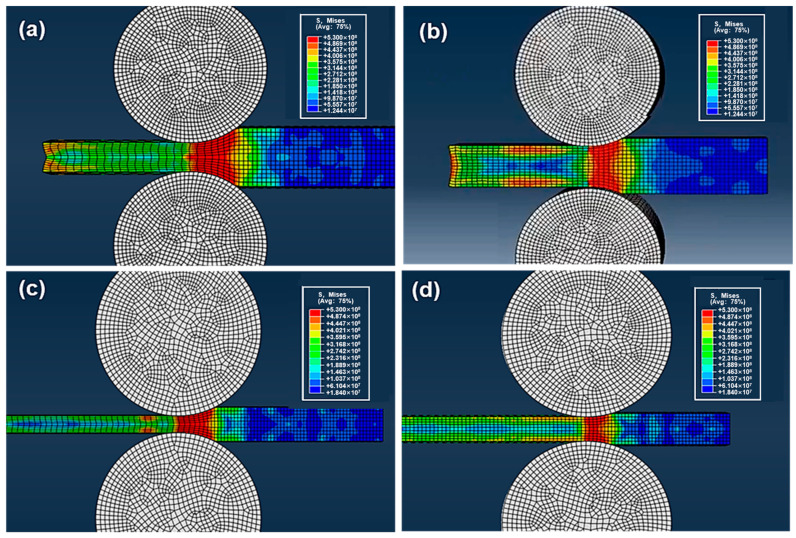
Simulation of the metal flow in rolling with different thickness aluminum plates and single reduction (**a**) 40 mm, 30%; (**b**) 40 mm, 15%; (**c**) 20 mm, 30%; (**d**) 20 mm, 15%.

**Figure 9 materials-14-02522-f009:**
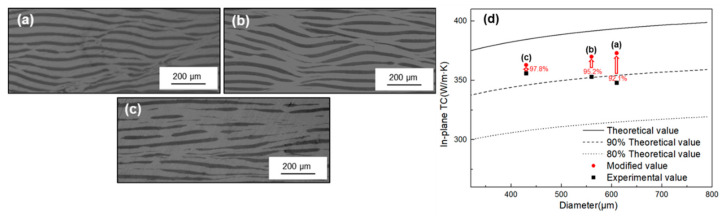
Morphology of composites with different initial thickness (**a**) 6 mm; (**b**) 8 mm; (**c**) 12 mm; (**d**) In-plane TC of (**a**–**c**).

**Figure 10 materials-14-02522-f010:**
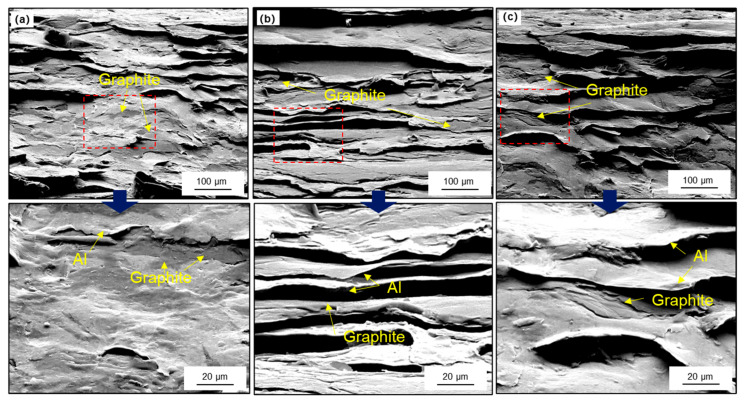
Tensile fracture morphology of composites with different total thickness (**a**) 6 mm; (**b**) 8 mm; (**c**) 12 mm.

**Figure 11 materials-14-02522-f011:**
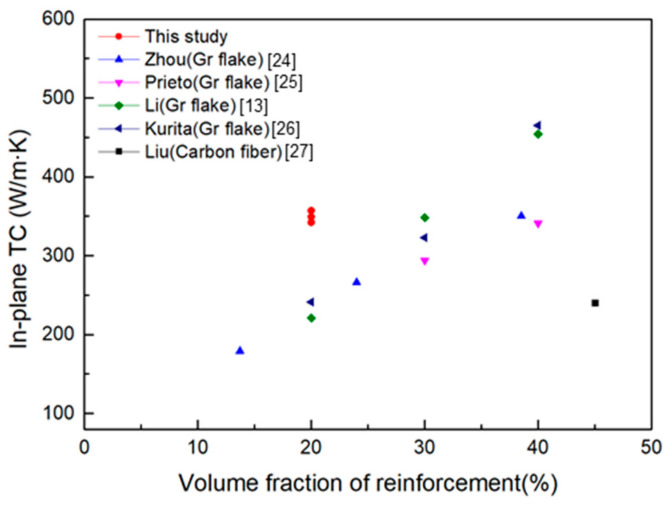
In-plane TC compared with other C/Al composites fabricated in the reference [24,25,26,27].

**Table 1 materials-14-02522-t001:** Properties of composites with different initial thickness.

Total Thickness (mm)	Average Size (μm)	Tensile Strength (MPa)	In-Plane TC (W/m·K)
8	612	56	343
12	557	63	351
16	439	77	354

## Data Availability

Data is contained within the article.

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
