# Peer review of "Study on Fabrication and Properties of Graphite/Al Composites by Hot Isostatic Pressing-Rolling Process"

_materials, 2021, doi:10.3390/ma14102522_

Round 1
Reviewer 1 Report
The manuscript is returned annotated with comments and suggestions
In this paper, ‘Fabrication and thermal conductivity of Graphite/Al composites by hot isostatic pressing-rolling process’ the authors report the fabrication of high thermal conductivity composite materials using graphite films and aluminum foil. The reported process to fabricate laminate material out of graphite film and Al foil by the hot isostatic pressing-rolling method will interest many readers involved in thermal management technology and related research. The heat transfer and thermal management research, including materials and techniques to remove high heat fluxes from power electronics, improve thermal system packaging as well as knowledge of multi-phase heat transfer fundamental mechanisms are hot topics and investigation in this area of research has continued to evolve. My impression is that using the rolling method reported by the authors in this paper, it is possible to fabricate and study the thermal properties of a host of isotropic and anisotropic materials to understand better the fundamental principles of heat transfers in such materials. Therefore, I recommend the publication of the paper in Materials after the authors must have made the following corrections and revision. Also, the authors must correct the many punctuation marks, commas and other typos and improve the presentation quality of their manuscript.
- Line 14: delete s from helps - the word should read 'help'
- Line 15: replace the comma with a full stop after ….. strength. The authors should see the comments and suggested corrections annotated in pdf file of their manuscript.
- 2.2 Preparation of the composite: Line 75 - Please describe how the stirring was performed
- Line 100: Figure 2b is not clear. The axes label is either missing or not included. The main profile is obscured, and it is difficult to read and understand the data presented in the text.
- Line 209: explain better how you determined the average size of the graphite.
- The authors should provide cross-section and magnified plane view SEM images in Figures 2, 3 7, and 8. This would help the readers to understand better the interlayer coupling between the graphite and Al, as well as clarify many of the claims made by the authors in the paper.

Reviewer 2 Report
-Title of the manuscript does not cover the whole parts of the study and needs to be modified.
-Abstract must contain brief and highlight achievements of research, not general discussions. Re-organization of the abstract is highly advised.
-The authors need significant improvement in English writing. There are many mistakes with regard to grammar (sentence structure, use of verbs, use of singular and plural forms, etc.) and spelling.
-The structure and subtitle of the different parts of the manuscript need to be completely modified.
-Figure 9b, Is not useful and should be excluded.
-In Figures 2,3,7, the different sections should be named on the micrographs.
-In Figures 2,3,7, The results of EDS analysis from different areas of graphite and aluminum layer and interface should be provided.
-In Figure 2,3,7, Higher magnification’s SEM images should be provided to show the structure of different parts of the composites.
-The relationship between the graphite and aluminum interface needs to be further explored. It is necessary to add some discussions on their interface adhesion.
-In the introduction, the fabrication of composites with other methods and their comparison should be discussed. For this purpose, it is recommended to review the following resources:
[a] Materials Science and Engineering: A 598, 162-173
[b] Ceramics International, 42, 2016, 15171-15176, DOI 10.1016/j.ceramint.2016.06.080
[c] Composites Part B: Engineering 125, 49-70
-Try to put some important discussions on the strengthening mechanisms of composite design.
-The outdated refs published before 2011 must be substituted with recent publications. Please consider Materials in your citing journals.
-The simulation part in figure 6 must be explained further. Also, the modeling part must be discussed in the experimental or as a separate section. Try to add all necessary criteria that you already considered during your approach.
Round 2
Reviewer 2 Report
The revised manuscript could be considered for publication in Materials.